# *Colletotrichum* Species Associated with Anthracnose in *Salix babylonica* in China

**DOI:** 10.3390/plants12081679

**Published:** 2023-04-17

**Authors:** Mengyu Zhang, Dewei Li, Yuanzhi Si, Yue Ju, Lihua Zhu

**Affiliations:** 1College of Forestry, Nanjing Forestry University, Nanjing 210037, China; mengyuzhang@njfu.edu.cn (M.Z.); siyuanzhi@njfu.edu.cn (Y.S.); juyue@njfu.edu.cn (Y.J.); 2Co-Innovation Center for Sustainable Forestry in Southern China, Nanjing 210037, China; 3The Connecticut Agricultural Experiment Station Valley Laboratory, Windsor, CT 06095, USA

**Keywords:** *Salix babylonica*, *Colletotrichum*, plant pathogen, pathogenicity

## Abstract

*Salix babylonica* L. is a popular ornamental tree species in China and widely cultivated in Asia, Europe, and North America. Anthracnose in *S. babylonica* poses a serious threat to its growth and reduces its medicinal properties. In 2021, a total of 55 *Colletotrichum* isolates were isolated from symptomatic leaves in three provinces in China. Phylogenetic analyses using six loci (ITS, *ACT*, *CHS-1*, *TUB2*, *CAL*, and *GAPDH*) and a morphological characterization of the 55 isolates showed that they belonged to four species of *Colletotrichum*, including *C. aenigma*, *C. fructicola*, *C. gloeosporioides* s.s., and *C. siamense*. Among them, *C. siamense* was the dominant species, and *C. gloeosporioides* s.s. was occasionally discovered from the host tissues. Pathogenicity tests revealed that all the isolates of the aforementioned species were pathogenic to the host, and there were significant differences in pathogenicity or virulence among these isolates. The information on the diversity of *Colletotrichum* spp. that causes *S. babylonica* anthracnose in China is new.

## 1. Introduction

*Colletotrichum* spp. are ones of the most important plant pathogens, saprobes and endophytes genera worldwide [1,2,3]. The fungal genus of *Colletotrichum* consists of 14 species or species complexes [4,5,6,7,8,9]. *Colletotrichum* pathogens often cause damage to roots, stems, leaves, flowers, fruits, and seedlings of trees, fruit trees, vegetables, flowers, medicinal plants, and field crops and can lead to plant wilting, anthracnose, fruit rot, leaf lesions, and other symptoms, causing serious economic losses [10,11]. Many species of *Colletotrichum* not only affect a wide range of host plants, but also have direct implications for human health [12,13,14]. Therefore, their accurate identification is critical because species differ in pathogenicity, fungicide sensitivity, and other factors affecting disease management in nurseries and seed orchards [15]. The taxonomy of the *Colletotrichum* species is quite complex [16]. The morphological identification of *Colletotrichum* species has long been difficult due to the plasticity of their morphological characteristics [10,17]. DNA sequences for identifying fungi are useful [18]. The internal transcribed spacer (ITS) region has been used as a barcoding locus for identifying fungi [19,20]. However, erroneous fungal identifications using ITS sequences have occurred [21,22]. Thus, it is difficult to identify fungi solely by the ITS region [21,22]. Therefore, in addition to the ITS region, other loci, such as *ACT* (actin), *CAL* (calmodulin), *CHS-1* (chitin synthase), *GAPDH* (glyceraldehyde-3-phosphate dehydrogenase), and *TUB2* (β-tubulin), have been applied to distinguish *Colletotrichum* species [1,22,23,24]. At present, multi-locus sequence data are widely used in the identification of *Colletotrichum* species [10,25,26,27,28,29].

*Salix babylonica* L. (*Salicaceae*) is distributed mostly in the northern hemisphere [30]. Since *S. babylonica* has a high ornamental value with its slender and graceful branches, it is widely planted by rivers and roadsides [31,32,33]. *Salix babylonica* also possesses a wide range of ecological characteristics, such as being easy to propagate, having a strong adaptability, and absorbing harmful gases, etc. [30,34,35,36,37]. In terms of utilization, *S. babylonica* has been increasingly employed in environmental restoration work and has shown promise for biofuel production and the phytoremediation of soil [37,38,39]. In addition, *S. babylonica* has an important medicinal value, the bark has astringent and tonic properties, and young twigs and catkins are antipyretic [40,41]. Modern medical research shows that the leaves of *S. babylonica* have good medicinal properties, such as relieving heat/fever, reducing inflammation, and detoxification [42]. However, *S. babylonica* is susceptible to diseases caused by phytopathogenic fungi. Anthracnose is one of the main diseases in *S. babylonica*. At the early stage of an anthracnos infection, there are small circular black spots on the leaves, which become irregular large spots. Finally, the whole leaf will wither. In 1997, anthracnose in *S. babylonica* was first reported in Greece [43]. The disease caused trees to lose their leaves repeatedly and seriously affected the ornamental value of the hosts. However, the morphological characteristics and taxonomy of *Colletotrichum* pathogens on *S. babylonica* have not been studied in detail.

From June to October 2021, anthracnose in *S. babylonica* occurred in three provinces in China. Therefore, this research study aimed to identify the *Colletotrichum* species causing anthracnose in *S. babylonica* based on morphological characteristics and multi-locus phylogenetic analyses and to determine the pathogenicity of the isolates with Koch’s postulates.

## 2. Results

### 2.1. Field Symptoms and Fungal Isolation

Anthracnose in *S. babylonica* was usually observed between June and October every year. The symptoms began as dark brown, irregular spots, and the centers were grayish white (Figure 1a–c). The spots gradually enlarged with time. Eventually the leaves withered and defoliated. Orange conidial masses often developed after the leaves were incubated in Petri dishes for 24 h with a high humidity (Figure 1d).

In this study, a total of six diseased sample batches were collected from six areas in the three provinces of China (Table 1). Thirty leaves were collected for each sample batch. A total of 55 *Colletotrichum* isolates were isolated according to their colony morphology on PDA and the ITS sequence data. Among these isolates, 12 isolates were from Suzhou, 10 isolates from Zibo, 10 isolates from Wuhan, and 23 isolates from Nanjing. Based on their ITS sequence data and colony characteristics on PDA, the isolates were divided into four types. Of these, 17 representative isolates were selected for further study and were sent to the China Forestry Culture Collection Center (CFCC).

### 2.2. Multi-Locus Phylogenetic Analyses

Seventeen representative isolates of *Colletotrichum* from different areas were selected for sequencing and analyses. The BLAST result of the ITS sequences showed that the 17 isolates belonged to the *C. gloeosporioides* species complex. They were analyzed using multi-locus sequences (ITS, *ACT*, *CHS-1*, *TUB2*, *CAL*, and *GAPDH*) and compared with 42 isolates of *Colletotrichum* (23 species), and *C. boninense* (CBS 123755) was used as the outgroup. A maximum likelihood estimation and Bayesian inference analyses with the concatenated sequences (ITS, *ACT*, *CHS-1*, *TUB2*, *CAL*, and *GAPDH*) identified the 17 isolates as *C. aenigma*, *C. fructicola*, *C. gloeosporioides* s.s., and *C. siamense* (Figure 2). Among these isolates, three isolates (HQ2-1, HQ2-6, and WH2-9) were in the same clade with *C. aenigma* with a bootstrap support value of 100; two isolates (SD1-6 and SD1-9) were in the same clade with *C. fructicola* with a bootstrap support value of 99; three isolates (WH2-4, NL1-7, and MXL1-7) were in the same clade with *C. gloeosporioides* s.s. with a bootstrap support value of 75; and nine isolates (YH2-2, YH2-3, YH2-5, YH2-6, WH2-7, NL1-10, NL1-13, MXL1-1, and MXL1-10) were grouped with *C. siamense* with high support values (ML/BI = 95/1).

### 2.3. Morphological Study

Based on the results of the phylogenetic analyses, the 17 *Colletotrichum* isolates characterized in this study belonged to four species: *C. aenigma* (three isolates), *C. fructicola* (three isolates), *C. gloeosporioides* (two isolates), and *C. siamense* (nine isolates). Representative isolates from each *Colletotrichum* species were selected to carry out detailed morphological descriptions.

#### 2.3.1. *Colletotrichum aenigma* B. Weir and P.R. Johnst (Figure 3) 

The colonies were white, and the aerial mycelium was white, dense, and cottony. In contrast to the colonies, the center was gray, and the margin was white. Orange conidial masses and ascomata were observed in the colonies. The colony growth rate on PDA was 13.2 mm/d. The acervuli were orange, elliptic, numerous, and pale to dark grey at the base. The conidiophores were hyaline to pale brown, smooth, septate, and sometimes branched. The conidiogenous cells were hyaline, cylindrical to ampulliform, smooth, thin-walled, (7.4–) 11.7–21.1 (–24.5) × (3.3–) 3.3–4.1 (–4.7) µm (mean ± SD = 16.4 ± 4.7 × 3.7 ± 0.4 µm (*n* = 30)), and with an L/W ratio = 4.4. The conidia were hyaline, aseptate, smooth, straight, subcylindrical, (12.6–) 15.1–16.3 (–16.7) × (4.7–) 5.3–6.1 (–6.3) µm (mean ± SD = 15.7 ± 0.6 × 5.7 ± 0.4 µm (*n* = 50)), with an L/W ratio = 2.7, and with a rounded end. The ascomata were brown to black, globose, and clustered. The asci were hyaline, clavate or fusiform, smooth, eight-spored, (52.5–) 58.1–71.5 (–78.5) × (7.4–) 10.8–13.8 (–16.4) µm (mean ± SD = 64.8 ± 6.7 × 12.3 ± 1.5 µm (*n* = 30)), and with an L/W ratio = 5.3. The ascospores were hyaline, aseptate, smooth, subcylindrical or ellipsoidal, slightly curved, uniseriate or biseriate, (14.5–) 16.5–20.7 (–20.3) × (3.9–) 4.2–5.2 (–5.4) µm (mean ± SD = 18.6 ± 2.1 × 4.7 ± 0.5 µm (*n* = 50)), and with an L/W ratio = 4.0. The appressoria were one-celled, ovoid or ellipsoidal, brown or dark brown, smooth, (6.5–) 7.3–9.1 (–9.8) × 4.6–7.0 (–7.0) µm (mean ± SD = 8.2 ± 0.9 × 5.8 ± 1.2 µm (*n* = 50)), and with an L/W ratio = 1.4. 

The specimens examined were as follows: (1) China, Hubei Province: Wuhan City, 30°43′10″ N, 114°31′59″ E, on the leaves of *Salix babylonica*, October 2021, Mengyu Zhang, culture WH2-9; (2) and China, Jiangsu Province: Suzhou City, 31°20′34″ N, 120°35′18″ E, on the leaves of *S. babylonica*, June 2021, Mengyu Zhang, cultures HQ2-1 and HQ2-6.

**Figure 3 plants-12-01679-f003:**
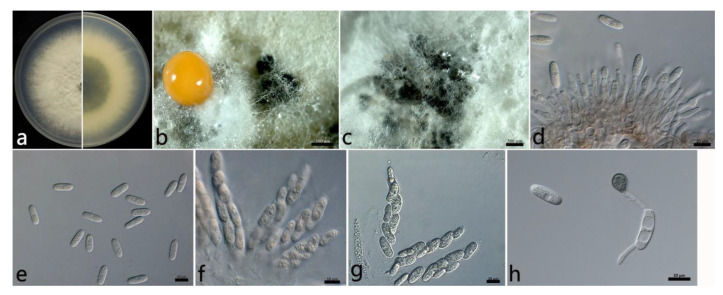
The morphological characteristics of *Colletotrichum aenigma* (WH2-9) isolated from anthracnose leaves of *Salix babylonica*. (**a**) Colony on PDA from above and below (5 d). (**b**) Conidial mass and ascomata (on PDA). (**c**) Ascomata (on PDA). (**d**) Conidiophores, conidiogenous cells, and conidia. (**e**) Conidia. (**f**) Asci and ascospores. (**g**) Ascospores. (**h**) Conidia and appressorium. Scale bars: (**b**,**c**) = 500 µm; (**d**–**h**) = 10 µm.

#### 2.3.2. *Colletotrichum fructicola* Prihastuti, L. Cai and K.D. Hyde (Figure 4)

The aerial mycelium was white to gray, dense, and cottony. In contrast to the colonies, the center was dark green, and the margin was white. Orange conidial masses and ascomata were observed in the center of the colonies. The colony growth rate on PDA was 13.6 mm/d. The acervuli were orange, elliptic, few, and pale to dark grey at the base. The conidiophores were hyaline to pale brown, smooth, septate, and sometimes branched. The conidiogenous cells were cylindrical to flask-shaped, hyaline, tapering towards the apex, smooth, thin-walled, (10.5–) 12.4–17.6 (–22.5) × (2.6–) 3.1–3.9 (–4.2) µm (mean ± SD = 15.0 ± 2.6 × 3.5 ± 0.4 µm (*n* = 50)), and with an L/W ratio = 4.3. The conidia were one-celled, aseptate, straight, subcylindrical, hyaline, (10.8–) 12.5–15.7 (–17.2) × (4.6–) 5.5–7.3 (–8.3) µm (mean ± SD = 14.1 ± 1.6 × 6.4 ± 0.9 µm (*n* = 50)), with an L/W ratio = 2.2, and with a rounded end. The ascomata were brown to black, round, and in clusters. The asci were hyaline, clavate, smooth, eight-spored, (40.3–) 40.5–52.7 (–55.0) × (8.1–) 8.6–11.4 (–13.0) µm (mean ± SD = 46.6 ± 6.1 × 10.0 ± 1.4 µm (*n* = 30)), and with an L/W ratio = 4.6. The ascospores were hyaline, aseptate, smooth, allantoid or ellipsoidal, curved, biseriate, (14.1–) 16.3–19.7 (–20.6) × (4.1–) 4.4–5.2 (–5.3) µm (mean ± SD = 18.0 ± 1.7 × 4.8 ± 0.4 µm (*n* = 50)), and with an L/W ratio = 3.8. The appressoria were one-celled, ovoid or ellipsoidal, brown or dark brown, smooth, (6.9–) 7.9–10.7 (–11.6) × (5.5–) 6.0–8.0 (–9.1) µm (mean ± SD = 9.3 ± 1.4 × 7.0 ± 1.0 µm (*n* = 50)), and with an L/W ratio = 1.3.

The specimens examined were as follows: China, Shandong Province: Zibo City, 36°37′58″ N, 117°53′43″ E, on the leaves of *Salix babylonica*, September 2021, Mengyu Zhang, cultures SD1-6 and SD1-9.

Notes: In this study, the conidia (12.5–15.7 × 5.5–7.3) and appressoria (7.9–10.7 × 6.0–8.0 (–9.1) µm) of the *C. fructicola* isolates were larger than those of the ex-type (ICMP 18581: 10.5–12.6 × 3.2–3.9 (–4.3) µm and 6.1–8.6 × 3.6–5.4 µm), respectively. For the sexual stage, the asci (40.5–52.7 × 8.6–11.4 µm) and ascospores (16.3–19.7 × 4.4–5.2 µm) were also larger than those of the ex-type (ICMP 18581: 34.2–48.2 × 7.0–8.2 µm and 10.5–13.3 × 3.0–3.7 (–4.0) µm), respectively. In the study by Prihastuti et al. [44], *C. fructicola* did not develop acervuli in PDA culture, but it developed acervuli in PDA in the present study (Figure 4b). The differences in morphology could be due to different hosts and should be further studied in the future. 

**Figure 4 plants-12-01679-f004:**
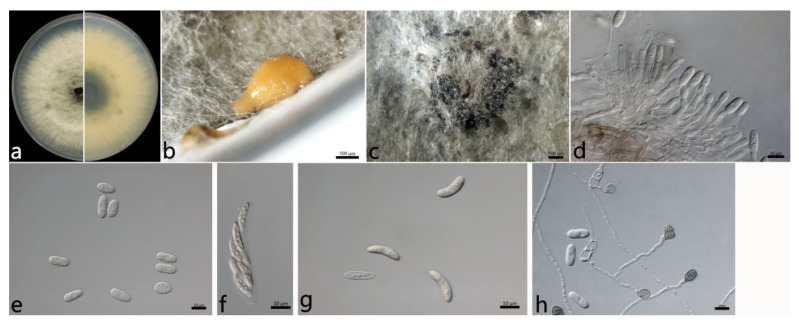
The morphological characteristics of *Colletotrichum fructicola* (SD1-6) isolated from anthracnose leaves of *Salix babylonica*. (**a**) Colony on PDA from above and below (5 d). (**b**) Conidial mass (on PDA). (**c**) Ascomata (on PDA). (**d**) Conidiophores, conidiogenous cells, and conidia. (**e**) Conidia. (**f**) Ascus. (**g**) Ascospores. (**h**) Conidia and appressoria. Scale bars: (**b**,**c**) = 500 µm; (**d**–**h**) = 10 µm.

#### 2.3.3. *Colletotrichum gloeosporioides* s.s. (Penz.) Penz. and Sacc (Figure 5)

The colonies on the PDA were white to grayish white at the center; in contrast, the center was dark green, and the margin was white. The aerial mycelium was white, dense, and cottony with a growth rate of 14.2 mm/d. Orange conidial masses were often observed in the center of the colonies. The acervuli were orange, elliptic, numerous, and pale to dark grey at the base. The conidiophores were hyaline to pale brown, smooth, septate, and rarely branched. The conidiogenous cells were cylindrical to flask-shaped, hyaline, tapering towards the apex, smooth, thin-walled, (6.5–) 9.4–18.8 (–21.0) × (3.3–) 3.4–4.2 (–4.5) µm (mean ± SD = 14.1 ± 4.7 × 3.8 ± 0.4 µm (*n* = 50)), and with an L/W ratio = 3.7. The conidia were hyaline, one-celled, aseptate, straight, subcylindrical with rounded ends, (12.1–) 14.0–16.0 (–16.9) × (5.7–) 6.2–7.0 (–7.3) µm (mean ± SD = 15.0 ± 1.0 × 6.6 ± 0.4 µm (*n* = 50)), and with an L/W ratio = 2.3. The appressoria were one-celled, ovoid or ellipsoidal, brown or dark brown, smooth, (7.2–) 7.8–9.6 (–10.7) × (5.9–) 5.8–7.2 (–8.4) µm (mean ± SD = 8.7 ± 0.9 × 6.5 ± 0.7 µm (*n* = 50)), and with an L/W ratio = 1.3.

The specimens examined were as follows: China, Jiangsu Province: Nanjing City, 32°5′10″ N, 118°49′13″ E and 32°3′2″ N, 118°50′26″ E, on the leaves of *Salix babylonica*, October 2021, Mengyu Zhang, cultures NL1-7 and MXL1-7; and China, Hubei Province: Wuhan City, 30°43′10″ N, 114°31′59″ E, on the leaves of *S. babylonica*, October 2021, Mengyu Zhang, culture WH2-4.

**Figure 5 plants-12-01679-f005:**
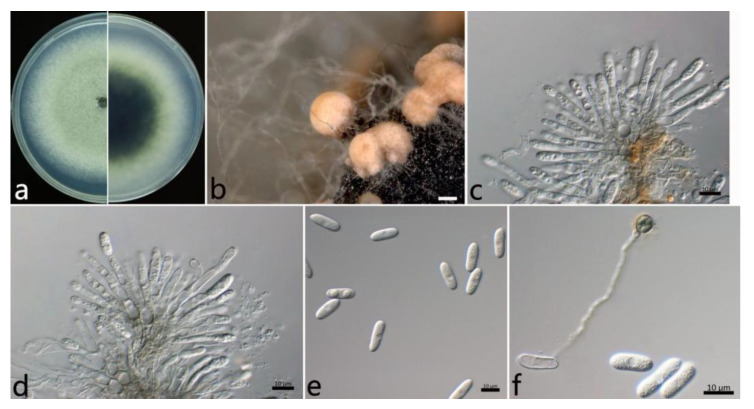
The morphological characteristics of *Colletotrichum gloeosporioides* s.s. (NL1-7) isolated from anthracnose leaves of *Salix babylonica*. (**a**) Colony on PDA from above and below (5 d). (**b**) Conidial masses (on PDA). (**c**,**d**) Conidiophores, conidiogenous cells, and conidia. (**e**) Conidia. (**f**) Conidia and appressorium. Scale bars: (**b**) = 500 µm; (**c**,**f**) = 10 µm.

#### 2.3.4. *Colletotrichum siamense* Prihastuti, L. Cai and K. D. Hyde (Figure 6)

The colonies on PDA were white to grayish white at the center. The aerial mycelium was abundant and cottony. Orange conidial masses were in the center of the colonies. The colony growth rate on PDA was 14.8 mm/d. The acervuli were orange, spherical or elliptical, numerous, and pale to dark grey at the base. The setae were dark brown, with two to three septates, thick-walled, straight, in groups, tapering toward the apices, and (85.4–) 75.9–111.1 (–117.6) µm (mean ± SD = 93.5 ± 17.6 μm (*n* = 30)). The conidiophores were hyaline to pale brown, septate, and branched. The conidiogenous cells were phialidic, hyaline, thin-walled, smooth, (9.6–) 10.8–17.4 (–20.0) × (2.3–) 2.9–3.9 (–4.6) µm (mean ± SD = 14.1 ± 3.3 × 3.4 ± 0.5 µm (*n* = 50)), and with an L/W ratio = 4.2. The conidia were one-celled, straight, subcylindrical, hyaline with a rounded end, (11.5–) 13.8–15.8 (–16.5) × (5.4–) 6.2–7.0 (–7.5) µm (mean ± SD = 14.8 ± 1.0 × 6.6 ± 0.4 µm (*n* = 50)), and with an L/W ratio = 2.3. The appressoria were one-celled, ovoid or ellipsoidal, brown or dark brown, smooth, (6.7–) 7.1–8.7 (–10.1) × (5.3–) 5.9–6.7 (–7.1) µm (mean ± SD = 7.9 ± 0.8 × 6.3 ± 0.4 µm (*n* = 50)), and with an L/W ratio = 1.3.

The specimens examined were as follows: China, Jiangsu Province: Suzhou City, 31°20′34″ N, 120°35′18″ E, on the leaves of *Salix babylonica*, June 2021, Mengyu Zhang, cultures YH2-2, YH2-3, YH2-5, and YH2-6; Nanjing City, 32°5′10″ N, 118°49′13″ E, and 32°3′2″ N, 118°50′26″ E, on the leaves of *S. babylonica*, October 2021, Mengyu Zhang, cultures NL1-10, NL1-13, MXL1-1, and MXL1-10; and China, Hubei Province: Wuhan City, 30°43′10″ N, 114°31′59″ E, on the leaves of *S. babylonica*, October 2021, Mengyu Zhang, culture WH2-7.

Notes: The ITS, *CHS*, and *TUB* sequences do not separate *C. siamense* from *C. fructicola*. However, these species are best distinguished using *CAL* sequencing and a multi-locus analysis. *Colletotrichum siamense* was first reported on the berries of *Coffea arabica* in Thailand [44]. Most previous studies have had difficulties distinguishing among *C. siamense*, *C. jasmini-sambac*, and *C. hymenocallidis* within the *C. gloeosporioides* complex [45,46]. However, later on, *C. jasmini-sambac* and *C. hymenocallidis* were demoted as synonyms of *C. siamense* [22].

**Figure 6 plants-12-01679-f006:**
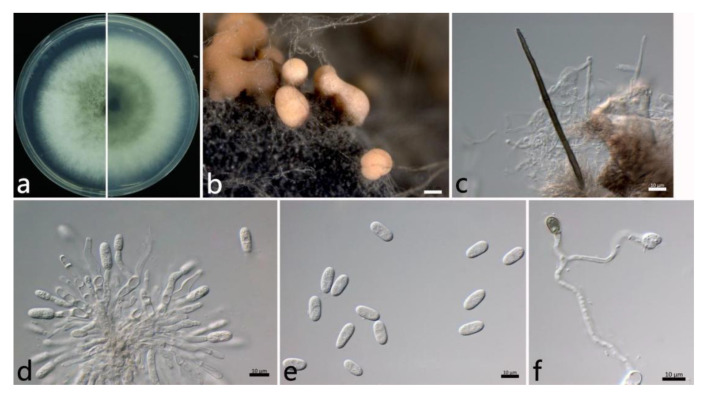
The morphological characteristics of *Colletotrichum siamense* (NL1-13) isolated from anthracnose leaves of *Salix babylonica*. (**a**) Colony on PDA from above and below (5 d). (**b**) Conidial masses (on PDA). (**c**) Seta. (**d**) Conidiophores, conidiogenous cells, and conidia. (**e**) Conidia. (**f**) appressoria. Scale bars: (**b**) = 200 µm; (**c**–**f**) = 10 µm.

### 2.4. Pathogenicity Tests

At 7 dpi, 17 representative isolates of the four *Colletotrichum* species developed dark brown lesion symptoms of anthracnose on the leaves of *S. babylonica* inoculated by a spore suspension. The infection incidence was 100%. No lesions were observed on the leaves of the control plants (Figure 7). However, different isolates had different levels of virulence, resulting in different lesions sizes. Among them, four out of the nine isolates of *C. siamense* had the most virulence, and *C. aenigma* had the least virulence (Table 2). The virulence within the same species of *C. siamense* and *C. gloeosporioides* s. s. varied significantly. The fungus was re-isolated from the infected tissues, and the morphology of the colony and the ITS sequence data matched the inocula. No fungi were isolated from the control leaves. The re-isolation rate was 100%. Thus, all 17 isolates were pathogens of anthracnose in *S. babylonica*.

## 3. Discussion

*Salix babylonica* is endemic in China and has a high ornamental value. Recently, anthracnose in *S. babylonica* has been discovered, seriously affecting the ecological value of *S. babylonica*. The identification of fungal pathogens is the most important first step for disease management [47]. In this study, we collected 55 isolates from six regions in three provinces where *S. babylonica* is grown and identified four known species of *Colletotrichum*.

Current identification systems for *Colletotrichum* species have included traditional morphological features, molecular phylogeny, and other traits [48]. However, these morphological features show plasticity under different conditions of growth (host, media, temperature, light regime, etc.), and some can be lost or change with repeated subculturing [22]. The conidia and ascospores developed on *S. babylonica* in this study are larger than those of the ex-type of *C. fructicola* (ICMP 18581) from *Coffea arabica*. Our morphological analyses also showed that the *Colletotrichum* species had the same sexual state characteristics under the same conditions. For example, *C. fructicola* and *C. aenigma* tend to develop asci and ascospores on PDA, resulting in the coexistence of sexual and asexual states. Thus, the identification of fungal pathogens in plants includes not only morphology but also multi-locus phylogenetic analyses [49,50]. For instance, Wang et al. [51] used three DNA sequences of ITS, *TUB2*, and *TEF1-α* to confirm a *Pestalotiopsis*-like species causing gray blight disease in tea plants in China. Poudel et al. [52] used ITS sequences to identify *Erysiphe fallax* causing powdery mildew on phasey beans in the United States. In this study, concatenated sequences of ITS, *ACT*, *CHS-1*, *TUB2*, *CAL*, and *GAPDH* were used to construct phylogenetic trees, and we identified the 17 isolates to be *C. aenigma*, *C. fructicola*, *C. gloeosporioides* s.s., and *C. siamense.*

The pathogenicity tests indicated pathogenic differences among the four species. *Colletotrichum siamense* had the highest virulence. In this study, *C. siamense* had the fastest colony growth rate on PDA, and correspondingly, it showed the highest virulence in the pathogenicity test. Secondly, the appressoria of *C. siamense* germinated easily. *Colletotrichum aenigma* had the slowest colony growth rate and showed the least virulence. The results indicated that the pathogenicity of the isolates was closely related to the colony growth rate and the appressorial germination rate. *Colletotrichum siamense* is an important pathogen that can infect many trees and fruits. For instance, *C. siamense* has been shown to cause anthracnose in pears, a number of host species in Proteaceae, and *Cunninghamia lanceolata* [25,53,54]. *Colletotrichum fructicola* was first reported in coffee berries from Thailand [44] and was later reported in *Pyrus pyrifolia* in Japan [22]. Subsequently, this species was widely recognized as the pathogen that caused pear anthracnose [55]. However, it can also infect other fruits, for instance, *Averrhoa carambola*, *Prunus sibirica*, and *Amygdalus persica* [56,57,58].

Based on pathogenicity test, *C. aenigma*, *C. fructicola*, *C. gloeosporioides* s.s., and *C. siamense* were identified as the pathogens of anthracnose in *S. babylonica*. Of them, *C. siamense* was the dominant species, and *C. gloeosporioides* s.s. was occasionally discovered from the host tissues. All of the isolates belong to the *C. gloeosporioides* species complex. The difference in the dominant species in the six regions may be due to different geographical locations, climates, host varieties, host health conditions, planting methods, and collection times [29]. Actually, many reports have shown that a host plant can be infected by several different *Colletotrichum* species. For example, chili is reported to be infected by *C. fioriniae*, *C. fructicola*, *C. gloeosporioides* s.s., *C. scovillei*, etc. [3]. Anthracnose in mango is caused by *C. asianum*, *C. fructicola*, *C. siamense*, *C. tropicale*, etc. [59]. Therefore, further studies are required to identify the host range and distribution of different *Colletotrichum* species.

It has been reported that *C. siamense*, *C. gloeosporioides* s.s., and *C. acutatum* can infect *S. babylonica* [33,60], but this study proved that *C. fructicola* and *C. aenigma* can also infect the leaves of *S. babylonica*. It is uncertain whether other *Colletotrichum* species can cause anthracnose in *S. babylonica*; extensive sampling in all distribution areas is required. In addition, the sensitivity of different *Colletotrichum* species to fungicides needs to be further studied. This is the first report on the diversity of *Colletotrichum* species associated with *S. babylonica* anthracnose worldwide. For controlling *S. babylonica* anthracnose effectively, these data will help us to select appropriate strategies for managing this disease.

## 4. Materials and Methods

### 4.1. Sample Collection and Fungi Isolation

From June to October 2021, the symptoms and pathogenesis of anthracnose in *S. babylonica* in different areas were assessed. Leaves with typical symptoms of anthracnose were randomly collected from six areas in three provinces (Jiangsu, Shandong, Hubei), China, and the samples (10 leaves/tree) were collected from three trees in each region. The samples were rinsed with running water for 10 min and dried in sterilized Petri dishes [61]. Small pieces of infected tissue (3–4 mm^2^) were surface-sterilized in 75% ethanol for 30 s followed by 1% NaClO for 90 s, rinsed three times in sterile water, dried on sterilized filter paper, plated on potato dextrose agar (PDA), and incubated at 25 °C in the dark [62,63]. Fungal growth was checked daily. Pure cultures were obtained by cutting hyphal tips and the monosporic isolation method [64]. All isolates were transferred to fresh PDA plates. The representative isolates were selected for further analyses and were sent to the China Forestry Culture Collection Center (CFCC).

### 4.2. DNA Extraction, PCR Amplification, and Sequencing

In order to obtain the genomic DNA of the strains, mycelium was harvested from colonies of fungal strains grown on PDA after 5 days of incubation at 25℃. Genomic DNA of 55 strains was extracted using the cetyltrimethylammonium bromide (CTAB) protocol [65]. Polymerase chain reaction (PCR) amplification was carried out on the extracted DNA. The internal transcribed spacer region (ITS), actin (*ACT*), chitin synthase (*CHS-1*), β-tubulin 2 (*TUB2*), calmodulin (*CAL*), and glyceraldehyde-3-phosphate dehydrogenase (*GAPDH*) loci were amplified using the primer pairs ITS1/ITS4 [66], ACT-512F/ACT-783R [67], CHS-79F/CHS-354R [67], T1/Bt2b [68,69], CL1C/CL2C [22], and GDF1/GDR1 [70], respectively (details of primers are given in Table 3). PCR mixture was performed in a total volume of 50 μL, containing 25 μL 2 × Taq Plus Master Mix, 19 μL double-distilled water, 2 μL primer-F, 2 μL primer-R, and 2 μL genomic DNA. The PCR conditions for ITS were 3 min at 94 °C; 30 cycles at 94 °C for 30 s; a 30 s cycle at 55 °C; a 45 s cycle at 72 °C; and then 10 min at 72 °C. The most suitable annealing temperatures differed for the other genes: ACT: 58 °C, CHS-1: 58 °C, TUB2: 55 °C, CAL: 55 °C, and GAPDH: 58 °C. For DNA sequencing, the PCR products were sent to Shanghai Sangon Biotechnology Co., Ltd., Shanghai, China.

### 4.3. Phylogenetic Analyses

The ITS, *ACT*, *CHS-1*, *TUB2*, *CAL*, and *GAPDH* sequences with high similarities to the genes/region sequences of *Colletotrichum* species in GenBank using BLAST were selected, and in total the sequences of 42 *Colletotrichum* isolates (23 species) were obtained from GenBank for phylogenetic analyses (Table 4). The sequences of *Colletotrichum boninense* (CBS 123755) were used as an outgroup. Nucleotide sequences of each gene/region of the selected isolates were aligned by the MAFFT ver. 7.313 [71]. The aligned sequences were edited using BioEdit version 7.0.9.0 [72]. Six locus sequences (ITS, *ACT*, *CHS-1*, *TUB2*, *CAL*, and *GAPDH*) were concatenated by PhyloSuite software [73]. After selecting the best model with ModelFinder [74], phylogenetic relationships were inferred using maximum likelihood (ML) estimation and Bayesian inference (BI). The ML analysis employed IQtree ver. 1.6.8 using the GTR+F+I+G4 model, with the bootstrapping method of 1000 replicates [75,76]. A bootstrap posed statistical support at ≥50%. BI analysis used the GTR+I+G+F model by MrBayes ver. 3.2.6, including 2 parallel runs and 2,000,000 generations [76]. Branches that received Bayesian posterior probabilities of 0.90 (BPP) were set as significantly supported. Phylogenetic trees were constructed with FigTree ver. 1.4.4.

### 4.4. Morphological Study

Morphological examinations focused on the colony characteristics, acervuli, conidiophores, conidiogenous cells, conidia, setae, appressoria, ascomata, asci, and ascospores of representative isolates that were randomly selected from each *Colletotrichum* species. Mycelial plugs (5 mm diam) from the margin of cultures were transferred to PDA and incubated at 25 °C in the dark. Colony characteristics were photographed with a Canon EOS M50 Mark II camera after 4 d, and colony diameters were measured daily to calculate the mycelial growth rates (mm/d). In order to induce appressorium formation, 10 µL of conidial suspension (10^6^ conidia/mL) was placed on a slide, placed inside plates containing a piece of moistened filter paper with sterile water, and then incubated at 25 °C in dark [77]. Measurements and morphological descriptions of acervuli, conidiophores, conidiogenous cells, conidia, setae, appressoria, ascomata, asci, and ascospores of the representative isolates were observed using a Zeiss Axio Imager A2m microscope (Carl Zeiss Microscopy, Oberkochen, Germany). Fifty individuals of per structure were measured for each isolate.

### 4.5. Pathogenicity Tests

Seventeen representative isolates of four *Colletotrichum* species were used for pathogenicity tests. Healthy 2-yr-old seedlings with 10 leaves per seedling were wound with a sterile needle and inoculated with conidial suspensions (10^6^ conidia/mL) in each leaf. The conidial suspensions were sprayed onto the wound. Control plants were treated with sterile water in the same way. Seedlings were covered with plastic bags after inoculation and maintained in a greenhouse at 25 ± 2 °C and 80% RH for seven days. The experiments were conducted three times, and each treatment had three replicates. Eventually 54 seedlings were used. Seven days after inoculation, the diameter of the lesion on the leaves was measured and the inoculated leaves were used for re-isolation.

## Figures and Tables

**Figure 1 plants-12-01679-f001:**
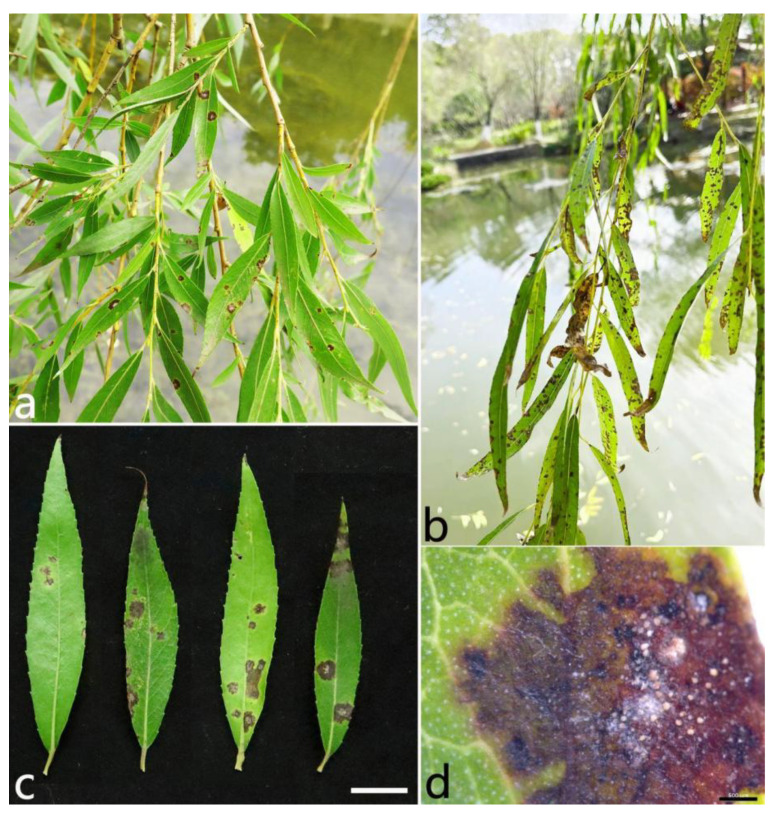
Symptoms of *Salix babylonica* anthracnose in the field. (**a**–**c**) Diseased leaves infected naturally. (**d**) Orange conidial masses after the leaves were incubated for 24 h under moist conditions. Scale bars: (**c**) = 1 cm; (**d**) = 500 µm.

**Figure 2 plants-12-01679-f002:**
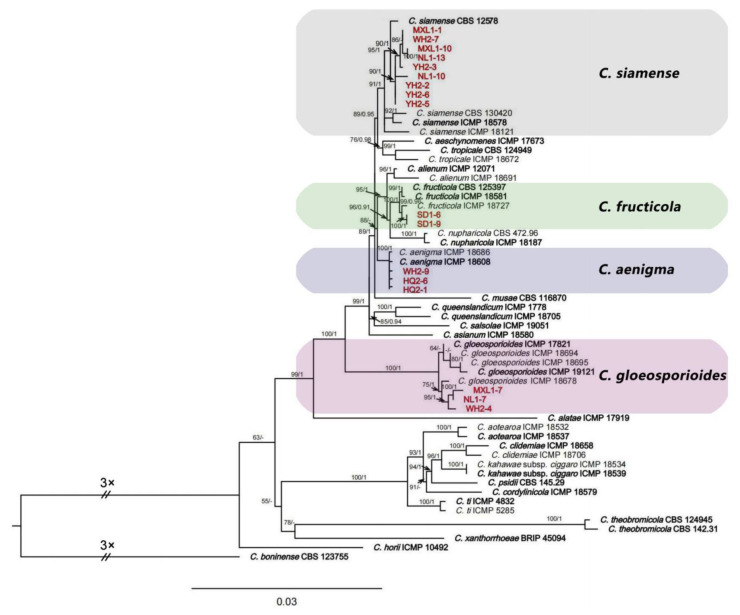
Phylogenetic relationship of *Colletotrichum* isolates (YH2-2, YH2-3, YH2-5, YH2-6, HQ2-1, HQ2-6, WH2-4, WH2-7, WH2-9, SD1-6, SD1-9, NL1-7, NL1-10, NL1-13, MXL1-1, MXL1-7, MXL1-10) from *Salix babylonica* with related taxa derived from the concatenated sequences of ITS, *ACT*, *CHS-1*, *TUB2*, *CAL*, and *GAPDH* loci using a maximum likelihood estimation and Bayesian inference analyses. Bootstrap support values (ML ≥ 50) and Bayesian posterior probability (PP ≥ 0.90) are shown at the nodes (ML/PP). *Colletotrichum boninense* (CBS 123755) is an outgroup. Bar = 0.03 substitutions per nucleotide position. Bold indicates ex-types. The red color text indicates strains of this study.

**Figure 7 plants-12-01679-f007:**
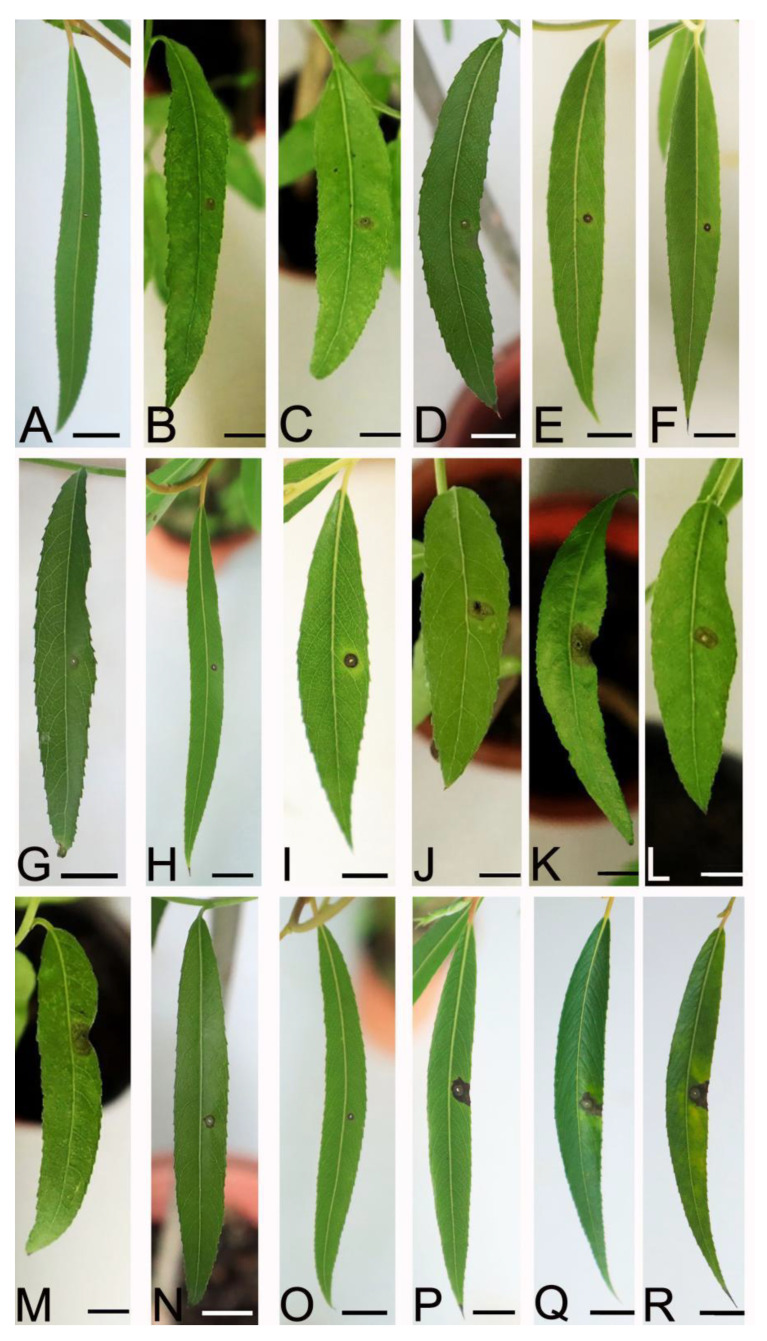
Symptoms on the leaves of *Salix babylonica* seedlings on day 7 after inoculation with conidial suspensions. (**A**) Control. (**B**–**D**) isolates HQ2-1, HQ2-6, and WH2-9 (*Colletotrichum aenigma*). (**E**,**F**) isolates SD1-6 and SD1-9 (*C. fructicola*). (**G**–**I**) isolates WH2-4, NL1-7, and MXL1-7 (*C. gloeosporioides*). (**J**–**R**) isolates YH2-2, YH2-3, YH2-5, YH2-6, WH2-7, NL1-10, NL1-13, MXL1-1, and MXL1-10 (*C. siamense*). Scale bars = 1 cm.

**Table 1 plants-12-01679-t001:** The sample list of *Colletotrichum* isolates collected from *Salix babylonica* in China.

Location	Host Tissue	Time	Latitude and Longitude	Number of Isolates
Suzhou, Jiangsu	leaf	2021.6.23	31°20′34″ N, 120°35′18″ E	6
Suzhou, Jiangsu	leaf	2021.6.23	31°18′30″ N, 120°34′41″ E	6
Zibo, Shandong	leaf	2021.9.11	36°37′58″ N, 117°53′43″ E	10
Wuhan, Hubei	leaf	2021.10.13	30°43′10″ N, 114°31′59″ E	10
Nanjing, Jiangsu	leaf	2021.10.20	32°5′10″ N, 118°49′13″ E	13
Nanjing, Jiangsu	leaf	2021.10.28	32°3′2″ N, 118°50′26″ E	10

**Table 2 plants-12-01679-t002:** The infection severity of representative *Colletotrichum* isolates on leaves of *Salix babylonica*.

No.	Species	Isolate	Lesion Length (mm)	No.	Species	Isolate	Lesion Length (mm)
1	*C. aenigma*	HQ2-1	1.4 ± 0.1 f	10	*C. siamense*	YH2-3	8.8 ± 0.1 a
2	*C. aenigma*	HQ2-6	1.6 ± 0.1 f	11	*C. siamense*	YH2-5	5.7 ± 0.1 c
3	*C. aenigma*	WH2-9	3.3 ± 0.2 e	12	*C. siamense*	YH2-6	5.6 ± 0.2 c
4	*C. fructicola*	SD1-6	3.2 ± 0.2 e	13	*C. siamense*	WH2-7	4.1 ± 0.3 d
5	*C. fructicola*	SD1-9	3.3 ± 0.1 e	14	*C. siamense*	NL1-10	1.4 ± 0.2 f
6	*C. gloeosporioides*	WH2-4	3.4 ± 0.2 e	15	*C. siamense*	NL1-13	8.3 ± 0.1 b
7	*C. gloeosporioides*	NL1-7	1.5 ± 0.3 f	16	*C. siamense*	MXL1-1	8.3 ± 0.2 b
8	*C. gloeosporioides*	MXL1-7	5.5 ± 0.2 c	17	*C. siamense*	MXL1-10	8.2 ± 0.2 b
9	*C. siamense*	YH2-2	5.6 ± 0.2 c				

Data were analyzed with SPSS Statistics 19.0 by one-way ANOVA, and means were compared using Duncan’s test at a significance level of *p* = 0.05. Letters indicate the significant difference at the *p* = 0.05 level.

**Table 3 plants-12-01679-t003:** PCR primers used for molecular characterization of *Colletotrichum* isolates.

Region	Primer	Direction	Sequence (5′–3′)	Tm (°C)
ITS	ITS1	Forward	TCCGTAGGTGAACCTGCGG	55
	ITS4	Reverse	TCCTCCGCTTATTGATATGC	
*ACT*	ACT-512F	Forward	ATGTGCAAGGCCGGTTTCGC	58
	ACT-783R	Reverse	TACGAGTCCTTCTGGCCCAT	
*CHS-1*	CHS-79F	Forward	TGGGGCAAGGATGCCTGGAAGAAG	58
	CHS-354R	Reverse	TGGAAGAACCATCTGTGAGAGTTG	
*TUB2*	T1	Forward	AACATGCGTGAGATTGTAAGT	55
	Bt2b	Reverse	ACCCTCAGTGTAGTGACCCTTGGC	
*CAL*	CL1C	Forward	GAATTCAAGGAGGCCTTCTC	55
	CL2C	Reverse	CTTCTGCATCATGAGCTGGAC	
*GAPDH*	GDF1	Forward	GCCGTCAACGACCCCTTCATTGA	58

**Table 4 plants-12-01679-t004:** A list of isolates of *Colletotrichum* spp. collected from *Salix babylonica* leaves in China as well as related taxa/isolates and their sequences used in this study.

Species	Culture *	Host	Country	GenBank Accession Number					
				ITS	*GAPDH*	*CAL*	*ACT*	*CHS-1*	*TUB2*
*C. aenigma*	ICMP 18608 *	*Persea americana*	Israel	JX010244	JX010044	JX009683	JX009443	JX009774	JX010389
	ICMP 18686	*Pyrus pyrifolia*	Japan	JX010243	JX009913	JX009684	JX009519	JX009789	JX010390
	HQ2-1	*S. babylonica*	China	OQ253546	OQ428578	OQ428572	OQ428569	OQ428575	OQ428581
	HQ2-6	*S. babylonica*	China	OQ243538	OQ428579	OQ428573	OQ428570	OQ428576	OQ428582
	WH2-9	*S. babylonica*	China	OQ253555	OQ428580	OQ428574	OQ428571	OQ428577	OQ428583
*C. aeschynomenes*	ICMP 17673 *	*Aeschynomene virginica*	USA	JX010176	JX009930	JX009721	JX009483	JX009799	JX010392
*C. alatae*	ICMP 17919 *	*Dioscorea alata*	India	JX010190	JX009990	JX009738	JX009471	JX009837	JX010383
*C. alienum*	ICMP 18691	*Persea americana*	Australia	JX010217	JX010018	JX009664	JX009580	JX009754	JX010385
	ICMP 12071 *	*Malus domestica*	New Zealand	JX010251	JX010028	JX009654	JX009572	JX009882	JX010411
*C. aotearoa*	ICMP 18532	*Vitex lucens*	New Zealand	JX010220	JX009906	JX009614	JX009544	JX009764	JX010421
	ICMP 18537 *	*Coprosma* sp.	New Zealand	JX010205	JX010005	JX009611	JX009564	JX009853	JX010420
*C. asianum*	ICMP 18580 *	*Coffea arabica*	Thailand	FJ972612	JX010053	FJ917506	JX009584	JX009867	JX010406
*C. boninense*	CBS 123755 *	*Crinum asiaticum* var. *sinicum*	Japan	JX010292	JX009905	--	JX009583	JX009827	--
*C. clidemiae*	ICMP 18706	*Vitis* sp.	USA	JX010274	JX009909	JX009639	JX009476	JX009777	JX010439
	ICMP 18658 *	*Clidemia hirta*	USA, Hawaii	JX010265	JX009989	JX009645	JX009537	JX009877	JX010438
*C. cordylinicola*	ICMP 18579 *	*Cordyline fruticosa*	Thailand	JX010226	JX009975	HM470238	HM470235	JX009864	JX010440
*C. fructicola*	ICMP 18581 *	*Coffea arabica*	Thailand	JX010165	JX010033	FJ917508	FJ907426	JX009866	JX010405
	ICMP 18727	*Fragaria × ananassa*	USA	JX010179	JX010035	JX009682	JX009565	JX009812	JX010394
	SD1-6	*S. babylonica*	China	OQ253556	OQ428565	OQ428561	OQ428559	OQ428563	OQ428567
	SD1-9	*S. babylonica*	China	OQ253557	OQ428566	OQ428562	OQ428560	OQ428564	OQ428568
*C. fructicola* (syn. *C. ignotum*)	CBS 125397 (*)	*Tetragastris panamensis*	Panama	JX010173	JX010032	JX009674	JX009581	JX009874	JX010409
*C. gloeosporioides*	ICMP 17821 *	*Citrus sinensis*	Italy	JX010152	JX010056	JX009731	JX009531	JX009818	JX010445
	ICMP 18694	*Mangifera indica*	South Africa	JX010155	JX009980	JX009729	JX009481	JX009796	--
	ICMP 18678	*Pueraria lobata*	USA	JX010150	JX010013	JX009733	JX009502	JX009790	--
	ICMP 18695	*Citrus* sp.	USA	JX010153	JX009979	JX009735	JX009494	JX009779	--
	WH2-4	*S. babylonica*	China	OQ243548	OQ428555	OQ428551	OQ428549	OQ428553	OQ428557
	NL1-7	*S. babylonica*	China	ON870951	ON858480	ON858478	ON858477	ON858479	ON858481
	MXL1-7	*S. babylonica*	China	OQ253571	OQ428556	OQ428552	OQ428550	OQ428554	OQ428558
*C. gloeosporioides* (syn. *Gloeosporium* *pedemontanum*)	ICMP 19121 (*)	*Citrus limon*	Italy	JX010148	JX010054	JX009745	JX009558	JX009903	--
*C. horii*	ICMP 10492 *	*Diospyros kaki*	Japan	GQ329690	GQ329681	JX009604	JX009438	JX009752	JX010450
*C. kahawae* subsp. *ciggaro*	ICMP 18539 *	*Olea europaea*	Australia	JX010230	JX009966	JX009635	JX009523	JX009800	JX010434
	ICMP 18534	*Kunzea ericoides*	New Zealand	JX010227	JX009904	JX009634	JX009473	JX009765	JX010427
*C. musae*	CBS 116870 *	*Musa* sp.	USA	JX010146	JX010050	JX009742	JX009433	JX009896	HQ596280
*C. nupharicola*	ICMP 18187 *	*Nuphar lutea* subsp. *polysepala*	USA	JX010187	JX009972	JX009663	JX009437	JX009835	JX010398
	CBS 472.96	*Nymphaea ordorata*	USA	JX010188	JX010031	JX009662	JX009582	JX009836	JX010399
*C. psidii*	CBS 145.29 *	*Psidium* sp.	Italy	JX010219	JX009967	JX009743	JX009515	JX009901	JX010443
*C. queenslandicum*	ICMP 1778 *	*Carica papaya*	Australia	JX010276	JX009934	JX009691	JX009447	JX009899	JX010414
	ICMP 18705	*Coffea* sp.	Fiji	JX010185	JX010036	JX009694	JX009490	JX009890	JX010412
*C. salsolae*	ICMP 19051 *	*Salsola tragus*	Hungary	JX010242	JX009916	JX009696	JX009562	JX009863	JX010403
*C. siamense*	ICMP 18121	*Dioscorea rotundata*	Nigeria	JX010245	JX009942	JX009715	JX009460	JX009845	JX010402
	ICMP 18578 *	*Coffea arabica*	Thailand	JX010171	JX009924	FJ917505	FJ907423	JX009865	JX010404
	YH2-2	*S. babylonica*	China	OQ243534	OQ428605	OQ428591	OQ428584	OQ428598	OQ428612
	YH2-3	*S. babylonica*	China	OQ253535	OQ428606	OQ428592	OQ428585	OQ428599	OQ428613
	YH2-5	*S. babylonica*	China	OQ253537	OQ428607	OQ428593	OQ428586	OQ428600	OQ428614
	YH2-6	*S. babylonica*	China	OQ253536	OQ428608	OQ428594	OQ428587	OQ428601	OQ428615
	WH2-7	*S. babylonica*	China	OQ253552	OQ428609	OQ428595	OQ428588	OQ428602	OQ428616
	NL1-10	*S. babylonica*	China	ON908707	ON858485	ON858483	ON858482	ON858484	ON858486
	NL1-13	*S. babylonica*	China	ON870949	ON858490	ON858488	ON858487	ON858489	ON858491
	MXL1-1	*S. babylonica*	China	OQ253561	OQ428610	OQ428596	OQ428589	OQ428603	OQ428617
	MXL1-10	*S. babylonica*	China	OQ253562	OQ428611	OQ428597	OQ428590	OQ428604	OQ428618
*C. siamense* (syn. *C. hymenocallidis*)	CBS 125378 (*)	*Hymenocallis americana*	China	JX010278	JX010019	JX009709	GQ856775	GQ856730	JX010410
*C. siamense* (syn. *C. jasmini-sambac*)	CBS 130420 (*)	*Jasminum sambac*	Vietnam	HM131511	HM131497	JX009713	HM131507	JX009895	JX010415
*C. theobromicola*	CBS 124945 *, ICMP 18649	*Theobroma cacao*	Panama	JX010294	JX010006	JX009591	JX009444	JX009869	JX010447
*C. theobromicola* (syn. *C. fragariae*)	CBS 142.31 (*)	*Fragaria × ananassa*	USA	JX010286	JX010024	JX009592	JX009516	JX009830	JX010373
*C. ti*	ICMP 5285	*Cordyline australis*	New Zealand	JX010267	JX009910	JX009650	JX009553	JX009897	JX010441
	ICMP 4832 *	*Cordyline* sp.	New Zealand	JX010269	JX009952	JX009649	JX009520	JX009898	JX010442
*C. tropicale*	ICMP 18672	*Litchi chinensis*	Japan	JX010275	JX010020	JX009722	JX009480	JX009826	JX010396
	CBS 124949 *	*Theobroma cacao*	Panama	JX010264	JX010007	JX009719	JX009489	JX009870	JX010407
*C. xanthorrhoeae*	BRIP 45094 *	*Xanthorrhoea preissii*	Australia	JX010261	JX009927	JX009653	JX009478	JX009823	JX010448

* indicates extype. BRIP: Plant Pathology Herbarium, Department of Employment, Economic, Development and Innovation, Queensland, Australia; CBS: Culture collection of the Westerdijk Fungal Biodiversity Institute, Utrecht, The Netherlands; ICMP: International Collection of Microorganisms from Plants, Auckland, New Zealand.

## Data Availability

All data generated or analyzed during this study are included in this article.

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
