# Peer review of "Colletotrichum Species Associated with Anthracnose in Salix babylonica in China"

_plants, 2023, doi:10.3390/plants12081679_

Round 1

Reviewer 1 Report

Dear colleagues,

It is a significant studies for phytopathogen in China and the quality of the figures is also very high. However, much improvment is required before the paper can be accepted. This manuscript was badly English written with many unclear sentences. You have learn to use simple and understandable English expression. Please improve your English writing and follow the Journal’s format. In addition, there are also big problems with the logic of the introduction and discussion, many of the conclusions are puzzling and unintelligible and don't know what you are trying to say.

Additional comments to improve the manuscript are noted in the upload PDF file.

Reviewer 2 Report

The study is relevant and interesting for the readers of the journal, as it provides new insights into the fungal diversity and disease management of anthracnose on S. babylonica and other willow species. However, the manuscript could be improved by addressing some major and minor issues before publication.

  • Major issues:
    • The title is too general and does not reflect the main objectives, methods, or results of the study. It also does not capture the novelty and significance of the research. The title should be revised to include the main method and result of the study, or to use a two-part title with a catchy phrase and a descriptive phrase. See the previous comments for some suggestions.
    • The introduction should provide more background information on Salix babylonica, its ornamental and medicinal uses and benefits, anthracnose disease, its symptoms and impacts on the host plant, Colletotrichum spp., their taxonomy and diversity, previous studies on Colletotrichum spp. associated with anthracnose on S. babylonica and other willow species, and the knowledge gaps and research questions that motivated this study. The introduction should also state the main aim or hypothesis of the study more clearly and explicitly.
    • The materials and methods section should provide more details on the sampling locations, dates, and methods, the isolation and purification of Colletotrichum isolates the materials and methods section should also cite any relevant references for the methods used or modified in this study.
    • The results section should present the results in a logical order that corresponds to the objectives or hypotheses of the study. The results section should also avoid repeating information that is already shown in tables or figures or vice versa. The results section should also use appropriate subheadings to organize the results according to different topics or aspects of the study.
    • The discussion section should also explain the implications and significance of the findings for the fungal diversity and disease management of anthracnose on S. babylonica and other willow species. The discussion section should also acknowledge any limitations or uncertainties of this study and suggest any future directions or recommendations for further research.
    • Some grammatical errors need to be corrected throughout the manuscript, such as verb tense consistency, subject-verb agreement, punctuation, capitalization, etc.
    • Some formatting errors need to be corrected throughout the manuscript, such as italicizing scientific names of genera and species, using

Round 2

Reviewer 2 Report

Manucript can be accepted.